# Analysis and Suppression of Laser Intensity Fluctuation in a Dual-Beam Optical Levitation System

**DOI:** 10.3390/mi13070984

**Published:** 2022-06-22

**Authors:** Xia Wang, Qi Zhu, Mengzhu Hu, Wenqiang Li, Xingfan Chen, Nan Li, Xunmin Zhu, Huizhu Hu

**Affiliations:** 1State Key Laboratory of Modern Optical Instrumentation, College of Optical Science and Engineering, Zhejiang University, Hangzhou 310027, China; xiawang@zju.edu.cn (X.W.); riggda@zju.edu.cn (Q.Z.); humengzhu@zju.edu.cn (M.H.); liwenqiang@zju.edu.cn (W.L.); mycotty@zju.edu.cn (X.C.); nanli@zju.edu.cn (N.L.); 2Quantum Sensing Center, Zhejiang Laboratory, Hangzhou 310000, China

**Keywords:** dual-beam optical trapping, optically levitated microsphere, laser intensity fluctuation suppression, relative intensity noise

## Abstract

Levitated micro-resonators in vacuums have attracted widespread attention due to their application potential in precision force sensing, acceleration sensing, mass measurement and gravitational wave sensing. The optically levitated microsphere in a counter-propagating dual-beam optical trap has been of particular interest because of its large measurement range and flexible manipulation. In this system, laser intensity fluctuation directly influences the trap stability and measurement sensitivity, which makes it a crucial factor in improving trapping performance. In this paper, a time-varying optical force (TVOF) model is established to characterize the influence of laser intensity fluctuation in a dual-beam optical trap. The model describes the relationship between the laser intensity fluctuation, optical force and the dynamic motion of the micro-sized sphere. In addition, an external laser intensity control method is proposed, which achieved a 16.9 dB laser power stability control at the relaxation oscillation frequency. The long-term laser intensity fluctuation was suppressed from 3% to 0.4% in a one-hour period. Experiments showed that the particle’s position detection sensitivity and the stability of the relaxation oscillation could be improved by laser intensity fluctuation suppression.

## 1. Introduction

Optical levitation systems have received increasing attention for their high mechanical quality. [1,2]. A levitated microsphere provides a candidate platform for precision acceleration and weak force measurement [3,4,5]. It has a wide range of potential applications in the detection non-Newtonian gravity, gravitational waves, and the Casimir force [6,7]. However, in the field of inertial navigation, long-term stable and movable precision acceleration is a critical limitation.

Researchers first revealed that the laser intensity fluctuation would affect the optical trap in atom traps [8]. The intensity noise could cause exponential heating [9,10]. Subsequently, electro-optical modulators and acousto-optic modulators were used to increase the trapping lifetime of atoms [11]. The laser intensity noise was found to cause the photon recoil phenomenon in the ultra-high vacuum of levitated nanospheres [12]. It had previously been observed that the laser intensity noise would cause instability in the levitation of atoms and nanospheres.

Optical traps with micron-sized particles have attracted significant interest in recent decades [13]. Compared with levitated nanospheres, levitated microspheres could achieve higher acceleration sensitivity. Recent works have shown that the detection sensitivity of an optical trap is affected by laser intensity noise and image noise [14,15]. Laser intensity fluctuation was found to exert a great influence on the acceleration detection sensitivity of an optical trap; however, systematic analyses of this phenomenon in a microsphere levitating system are lacking.

In this work, we propose a time-varying optical force (TVOF) model of particle movement in a dual-beam optical trap. The TVOF model describes the relationship between the laser intensity fluctuation, optical force and the dynamic properties of the particle. Then, the main factors influencing the stability of dual-beam optical trap are analyzed. An external laser intensity control method is proposed to suppress the effect of laser intensity fluctuation. Finally, an experiment is performed, and the results show that the particle’s position detection sensitivity and the stability of the relaxation oscillation are improved by laser intensity fluctuation suppression.

## 2. The TVOF Model

### 2.1. Optical Forces in a Dual-Beam Optical Trap

A particle trapped by two counter-propagating beams experiences a scattering force and a gradient force, as shown in Figure 1. The main factors affecting the stability of dual-beam optical traps are the alignment of the dual-beam and the fluctuation of the optical force. The motion of a particle in air is affected by the air molecules and the optical force. Through the accurate adjustment, the alignment accuracy of a dual-beam can be ensured. The influence of air molecules can be controlled by changing the vacuum pressure. The optical force is in proportion to the beam intensity and is mainly affected by the intensity fluctuation.

As the particle radius discussed here is larger than the wavelength, a geometrical optics method could be applied in the analysis of the optical force affecting the particle. Based on the conservation of momentum and the geometrical optics [16], the optical force of a single beam can be derived. The gravity force of the particle is neglected, so $F_{x}=F_{y}$.
(1)$$\left\{\begin{array}{l} dF_{z}=\frac{n_{1}}{c}(1+R_{e}\cos2\alpha_{i}-T_{e}{}^{2}\frac{\cos(2\alpha_{i}-2\alpha_{r})+R_{e}\cos2\alpha_{i}}{1+R_{e}{}^{2}+2R_{e}\cos2\alpha_{r}})dP\\ dF_{x}=dF_{y}=\frac{n_{1}}{c}(-1+R_{e}\sin2\alpha_{i}+T_{e}{}^{2}\frac{\sin(2\alpha_{i}-2\alpha_{r})+R_{e}\sin2\alpha_{i}}{1+R_{e}{}^{2}+2R_{e}\cos2\alpha_{r}})dP \end{array}\right.$$
where $F_{z}$ represents the scattering force, $F_{x}$ represents the gradient force, $n_{1}$ represents the refractive index of the environmental medium, c represents the vacuum speed of light, dP represents the laser intensity of a single ray, $R_{e}$ and $T_{e}$ are Fresnel reflection and transmission coefficients, and $\alpha_{i}$ and $\alpha_{r}$ represent the incidence angle and transmission angle, respectively.

### 2.2. Effect of Laser Intensity Fluctuation on the Trap Stability

The intensity imbalance of the two lasers is an important cause of the instability of the particle in a dual-beam optical trap. Notably, perfect balance is almost impossible to achieve in real systems. The intensity difference in the dual beams can be caused by adjustment error, differences in the paths and long-term system drift. The intensities of two beams are $P_{1}$ and $P_{2}$. Then, $P_{2}=q_{r}P_{1}$, where $q_{r}$ is the intensity mismatch ratio.

The optical forces exerted on an SiO_2_ particle with a size of 10 µm are calculated for different $q_{r}$ values, as shown in Figure 2.

As shown in Figure 2, the optical forces and trap stiffness change with $q_{r}$. In the Z axis, with the decrease of $q_{r}$, the equilibrium position moves to the weak intensity side of the dual-beam and disappears when $q_{r}=0.8115$. The trap stiffness decreases with the increase in $q_{r}$. Along the *X* axis, the trapping potential well ($\mathrm{TPW}=F_{\mathrm{lbm}}/\mathrm{ELR}$) decreases with $q_{r}$. When $q_{r}$ increases, the trap stiffness linearly decreases.

The total scattering force in a dual-beam optical trap is the force difference between the scattering forces of the two beams. The total gradient force is the sum of the gradient forces of the two beams, and the intensity mismatch ratio has a greater effect along the *Z* axis.

We can conclude that $q_{r}$ should be larger than 0.8115 to obtain stable trapping, and that the trap stiffness fluctuates with the variation in $q_{r}$.

### 2.3. Effect of Laser Intensity Fluctuation on Particle Movement

#### 2.3.1. Statistical Analysis of the Synchronous Laser Intensity Fluctuations

In a dual-beam optical trap, the laser intensity fluctuations of the laser source cause the synchronous intensity change in both beams. In Figure 3a,b, ELR remains unchanged with the increase in $q_{lfr}$ when $q_{r}$ is fixed. $F_{lbm}$ and TPW increase with $q_{lfr}$. In Figure 3c, d, we can see that when $q_{r}$ is fixed, the stiffness obviously decreases linearly with the increase in $q_{lfr}$ along the *X* and *Z* axes. The optical force $F_{opti}\left(t\right)$ could be simplified as shown in Formula (2):(2)$$F_{opti}\;\left(t\right)=\left(k_{i}+\Delta k_{i}\left(t\right)\right)x_{i}=\left(k_{i}+\frac{\Delta P(t)}{\alpha_{ri}\bar{P}}\right)x_{i},\;i=X,\;Y\;\mathrm{and}\;Z$$
where $k_{i}$ and $\Delta k_{i}(t)$ represent the stiffness and the variation of the stiffness, respectively, along the *X*, *Y* and *Z* axes; $\alpha_{ri}$ denotes a constant coefficient; and $x_{i}$ is the center-of-mass displacement of the microsphere along the *X*, *Y* and *Z* axes.

#### 2.3.2. Dynamic Analysis of Laser Intensity Fluctuation

Here, the movement of particle affected by laser intensity fluctuation under thermal equilibrium state is analyzed. The main factors are laser optical force $F_{opt}$, molecular collision force $F_{therm}$, gravity $F_{g}$, and damping term $F_{visc}$. For a micron-sized particle at thermal equilibrium state, the recoil energy $\hbar^{2}k^{2}/2m$ and gravity term can be neglected [12]. The dynamic properties of the trapped particle in the vacuum [17] are derived by Formula (3):(3)$$m\frac{d^{2}x}{dt^{2}}+\left(k_{i}+\frac{\Delta P(t)}{\alpha_{ri}\bar{P}}\right)x+\gamma \frac{dx}{dt}=\left(\xi_{i}\left(t\right)\sqrt{{2k}_{B}T_{0}\gamma }\right)$$
where x, m, $T_{0}$, $k_{B}$, and $\xi_{i}\left(t\right)$ are the center-of-mass displacement, the mass, the environmental temperature, Boltzmann’s constant and white noise, respectively; and γ represents Stokes friction coefficient.

Laser intensity fluctuation includes long-term intensity fluctuation and relative intensity noise (RIN) [18]. Long-term intensity drift is caused by the mechanical vibration of optical components and long-term drift. Meanwhile, RIN is caused by the fluctuation of the laser source’s gain medium and the pump current [19,20].

Long-term intensity fluctuation changes the mean stiffness of the trap. The resonant frequency of the particle is $f_{0}=\omega_{0}/2\pi =\sqrt{k/m}/2\pi$. We used Runge-Kutta methods to solve Formula (3) and obtain the center-of-mass displacement x. The power spectral density (PSD) is thus calculated from the center-of-mass displacement. When the force or acceleration acting on the microsphere changes, it influences the center-of-mass motion and is reflected in the PSD [21]. As shown in Figure 4a, the PSD of the center-of-mass displacement and the resonant frequency change with stiffness. As can be seen in Figure 4b, with the occurrence of RIN, the amplitude of the PSD increases. The RIN introduces heating into the movement of the particle. At around 310 Hz, the PSD of the red line is below the black line, which indicates the presence of a cooling effect. We consider that the cooling effect is due to the force of the molecular collision, and RIN are occasionally opposite at this frequency.

From the above, we can conclude that long-term intensity fluctuation changes the resonant frequency and affects the detection stability. The RIN heats the movement of the particle in the dual-beam optical trap and affects the position sensitivity.

## 3. Laser Intensity Fluctuation Suppression

According to the above TVOF model, the laser intensity fluctuation affects the stability of the dual-beam optical trap and the position detection. Therefore, an external laser intensity control method based on a digital signal control module is proposed. The method suppresses long-term intensity fluctuation and RIN in real time. The control method is realized based on the principle of an acousto-optic modulator (AOM) and by using its 1st order Bragg diffraction light. The high-frequency scattershot noise is filtered. The external laser intensity control method is shown in Figure 5, which includes five steps as follows:(a)Set the system initial parameter and AD sampling laser intensive data. The sampling frequency is determined by the particle resonance frequency and the RIN relaxation oscillation, and a sliding smoothing filter is used to eliminate the influence of an abnormal transient singularity signal.(b)The FPGA storage short-time sampling of data is performed in real-time. The RIN data are acquired from real-time sampling, and the mean value in short-time is obtained.(c)Comparing the storage data and real-time data, the suppression effect and control speed is calculated. The factor values to control the RIN data are auto adjusted.(d)The long-term drift factor is adjusted by the history storage and setting parameter.(e)All parameters acting on the final control output are compared with the corresponding value threshold. The output signal controls the AOM to suppress the laser intensity in long-term intensity fluctuation and RIN.

## 4. Experiment

The dual-beam optical trap contains a laser intensity fluctuation suppression subsystem, a levitation subsystem and a detection subsystem, as shown in Figure 6.

In the levitation subsystem, the laser source is 1064 nm, with a maximum laser power output of 5 W. The source beam is divided into a p-polarized beam (LB1) and an s-polarized beam (LB2) by a polarization beam splitter. Two objectives are used to focus the beams, and their numerical aperture is 0.15. The highly focused orthogonally polarized lights are used to avoid diffraction and form the dual-beam optical trap. The input beam diameter is 2 mm, which is much smaller than the lens aperture. The effective numerical apertures of both beams are approximately 0.07. The waist radius of the focus $\omega_{0}$ is approximately 4.8 µm, and the Rayleigh distance $Z_{0}$ is 65 µm. The beam intensities are controlled by the combination of a half-wave plate and a polarization beam splitter cube. The power of each trapping beam before it enters the vacuum chamber is 1 W. The two optical axes of collimated beams are finely adjusted to be parallel by an M05-X Mirror Mount. A 5 µm pinhole and piezo inertia actuators (8301NF, Newport Inc., Irvine, CA, USA) with a minimum step size of dm = 30 nm are used to align the beams. It is ensured that the two beams are focused on the same point [22]. The levitated particle is an SiO_2_ sphere that is 3 µm in diameter. The air pressure in the vacuum chamber is 10 mbar, and the room temperature is 293 K.

The feedback is implemented with a laser intensity fluctuation suppression subsystem. The laser intensity fluctuation signal is detected by a high-speed photodiode and then processed by an FPGA digital system. The FPGA digital system realizes the external laser intensity control method and modulates the RF power in an AOM to suppress the laser intensity fluctuation in real-time.

In order to monitor the real-time position of a particle, a detection subsystem is built using a photoelectric balance detector to detect the motion of a particle along the *X*, *Y* and *Z* axes. In the detection subsystem, the particle acts as a spherical lens; the movement of particle deflects the scattered beam along the *X* and *Y* axes. D-shape mirrors are carefully adjusted to ensure that they can divide beams along the axes. The motion of a trapped particle along the *Z*-axis changes the divergence angle of the scattering light, which alters the waist of incident beams. The differential detection method is used to detect the motion along the *Z* axis. The output voltages of PBDX, PBDY and PBDZ are linearly related to the particle’s motion along the *X*, *Y* and *Z* axes, and their linear coefficients are $\beta_{x}$, $\beta_{y}$ and $\beta_{z}$, respectively.

The comparison of laser intensity fluctuation between the on and off state of the external laser intensity control method is shown in Figure 7. It can be seen that the suppression effect reaches 16.9 dB at relaxation oscillation. The suppression effect frequency range is 0 Hz~72 kHz. The one-hour long time stability is raised from 3.069% to 0.4425%.

Next, the mismatch ratio of LB1 and LB2 is researched. When the laser intensity of LB1 is 1 W, the mismatch ratios are 95% and 88%, respectively. The particle is lost when the mismatch ratio is below 80%. The particle is captured at an 88% mismatch ratio at 10 mbar, as shown in Figure 8a. It can be seen that the experimental results agree with the theory.

As seen in Figure 8, after the external laser intensity control, the position detection sensitivity is improved in both the low and high frequency range. The detection sensitivity of the Z-axis is partially improved from 29.51 $\mathrm{pm}/\sqrt{Hz}$ to 17.15 $\mathrm{pm}/\sqrt{Hz}$ in the 8 kHz to 18 kHz range.

Subsequently, trap stiffness can be measured through the resonance frequency of each axis. After obtaining the resonant frequencies $f_{0}$ in 1 h with 248 sampling points, the results show that the resonant frequencies have a Gaussian distribution, as shown in Figure 9. The fitted full-width half maximum (FWHM) of the *X* and *Z* axes changes from 210 and 207 to 164 and 140, respectively, after the long-term suppression. Their distribution is concentrated after suppression; the FWHM is decreased by 22% and 32%, respectively.

The experimental results reveal that the position detection stability and sensitivity are improved through the fluctuation of laser intensity noise suppression. The long-term intensity fluctuation and RIN suppression effects are consistent with the theoretical analysis.

## 5. Conclusions

Herein, we characterized the influence of laser intensity fluctuation in a dual-beam optical trap. A TVOF model was established to describe the relationship between laser intensity fluctuation, optical force and the dynamic properties of a micro-sized particle. Then, laser intensity noise suppression based on an external laser intensity control method was proposed. Finally, 16.9 dB laser power stable control was achieved at the relaxation oscillation. The experimental results, which matched the simulation, showed that the particle could not be captured when the mismatch ratio was below 80%. After RIN suppression, the position detection sensitivity was partially improved from 29.51 $\mathrm{pm}/\sqrt{Hz}$ to 17.15$\mathrm{pm}/\sqrt{Hz}$ in the 8 kHz to 18 kHz range. The long-term stability was improved from 3% to 0.4%. Following suppression, the FWHM of the resonant frequency distribution decreased by 22% and 32%. The system stiffness distribution was concentrated through long-term intensity fluctuation suppression. Overall, our method effectively suppressed the intensity fluctuations in the dual-beam optical trap and provided an effective solution for improving system accuracy, which may pave the way for novel high sensitivity accelerometers. This method can be applicable to particles ranging from nanospheres to microspheres. In follow-up research, we will aim to analyze the laser intensity fluctuation effect in optical traps with particles of other sizes and in optical traps that are in a non-equilibrium state. In addition, the influence of the laser intensity fluctuation caused by the cooling light and other noises in the system at a high vacuum will be further analyzed to achieve higher sensitivity.

## Figures and Tables

**Figure 1 micromachines-13-00984-f001:**
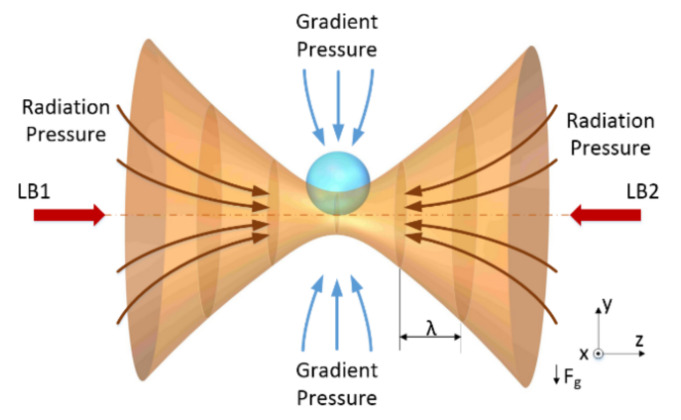
The dual-beam optical trap uses two strongly focused beams for trapping particles. The two beams named LB1 and  LB2.  The intensity gradients in the two converging beams draw particles toward the focus, whereas the radiation pressures of the two beams tend to blow them toward the optical axis. Under conditions where the two beams have the same intensity, the beams are coaxial and focused on the same point. A particle can be trapped in three dimensions, near the focal point.

**Figure 2 micromachines-13-00984-f002:**
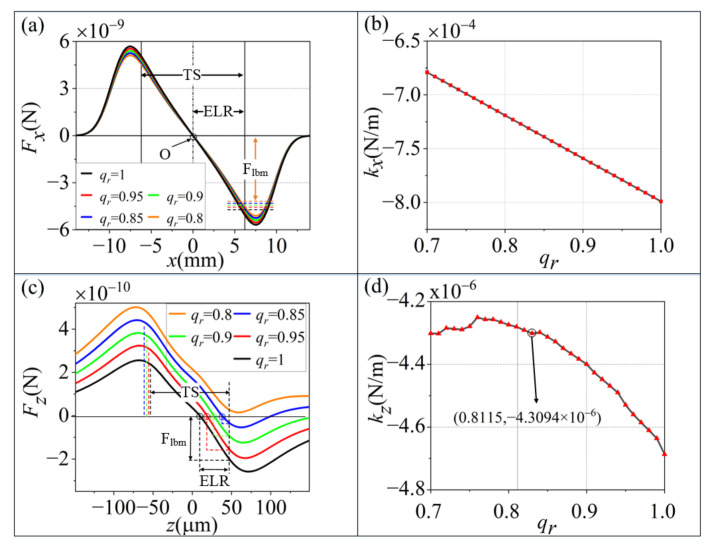
Optical forces on a sphere in a Gaussian dual-beam trap when $q_{r}$ changes from 0.8 to 1. O, optical force equilibrium position; TS, trapping scope; ELR, effective linear region; and $F_{\mathrm{lbm}}$, the minimum absolute force at the trapping scope borders. (**a**) is the relationships of gradient force $F_{x}$ and different $q_{r}$ values along the *X* axis. (**c**) is the relationships of scattering force $F_{x}$ and different $q_{r}$ values along the *Z* axis. (**b,d**) are the relationships of the trap stiffness and $q_{r}$ along the *X* and *Z* axes, respectively.

**Figure 3 micromachines-13-00984-f003:**
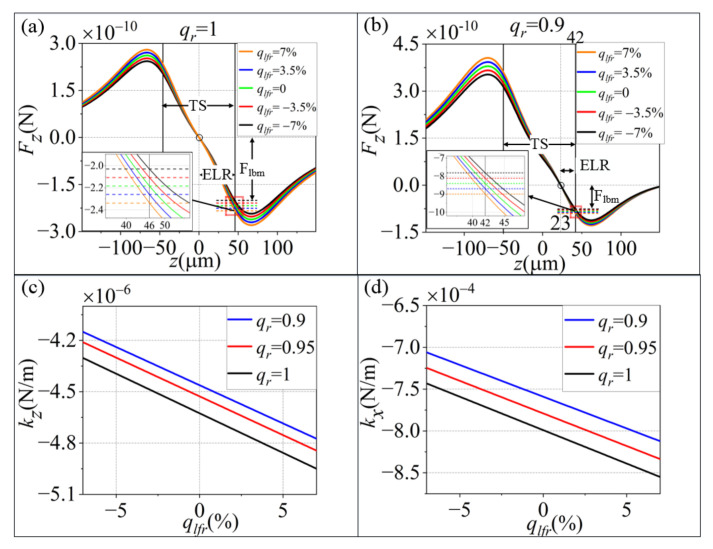
The optical force and stiffness fluctuation with the laser intensity fluctuation at different fixed $q_{r}$ values. (**a**) $q_{r}$ = 1 and (**b**) $q_{r}$ = 0.9 along the *Z*-axis; (**c**,**d**) are the relationships of laser intensity fluctuation and stiffness at different fixed $q_{r}$ values along the *Z* and *X* axes, respectively. Here, laser intensity $P(t)=\Delta P(t)+\bar{P}$, where $\bar{P}$ denotes the mean intensity, $q_{lfr}$ is the synchronous intensity change rate, and $q_{lfr}=\Delta P/\bar{P}$.

**Figure 4 micromachines-13-00984-f004:**
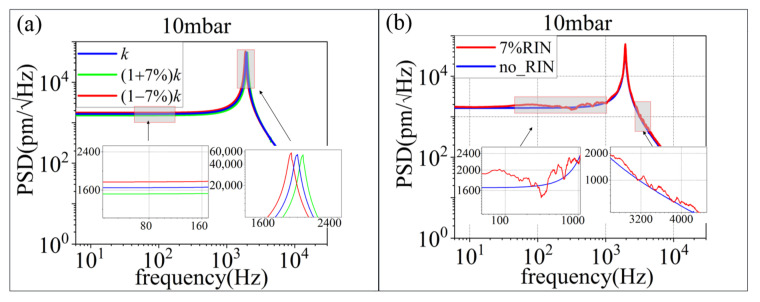
The simulation results of particle movement along the laser intensity fluctuation at 10 mbar. (**a**) Long-term intensity fluctuation (**b**) RIN.

**Figure 5 micromachines-13-00984-f005:**
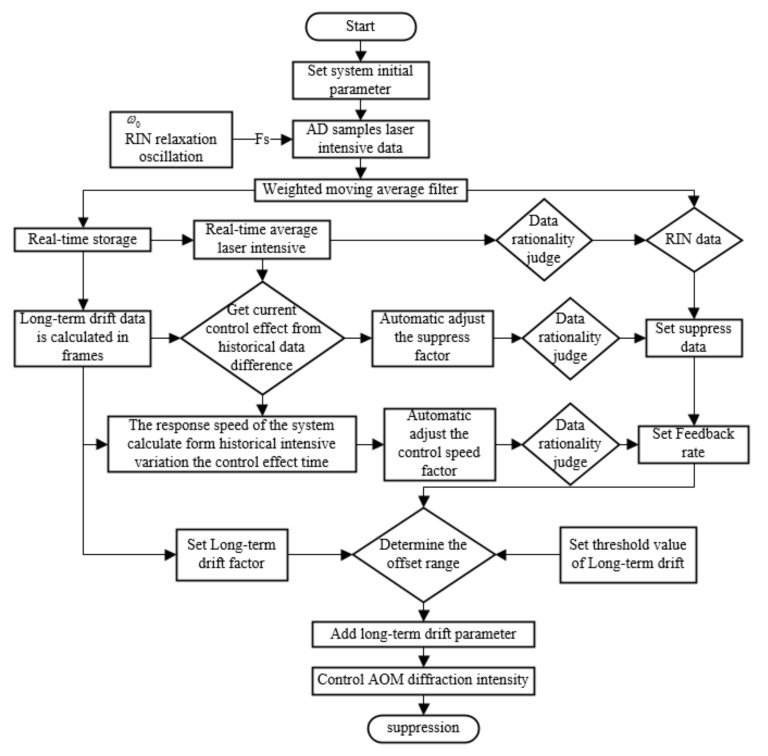
Suppression process of the external laser intensity control method.

**Figure 6 micromachines-13-00984-f006:**
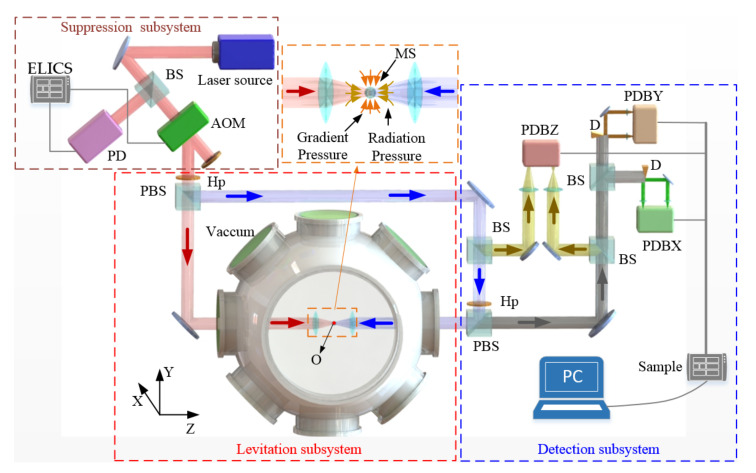
Schematic of a dual-beam optical trap with laser intensity fluctuation suppression. The abbreviations are: BS, beam splitter; PBS, polarization beam splitter; PD, high-speed photodiode; ELICS, external laser power control system; Hp, half-wave plate; MS, microsphere; D, D shape mirror; PDBX,PDBY and PDBZ, photoelectric balance detector to detect the motion of MS along the *X*, *Y*, *Z* axes, respectively.

**Figure 7 micromachines-13-00984-f007:**
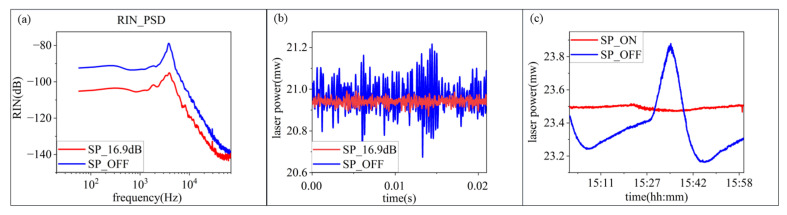
Suppression results of laser source before and after 16.9 dB suppression. (**a**) The PSD of RIN. (**b**) The time domain waveform. (**c**) The long-term intensity fluctuation.

**Figure 8 micromachines-13-00984-f008:**
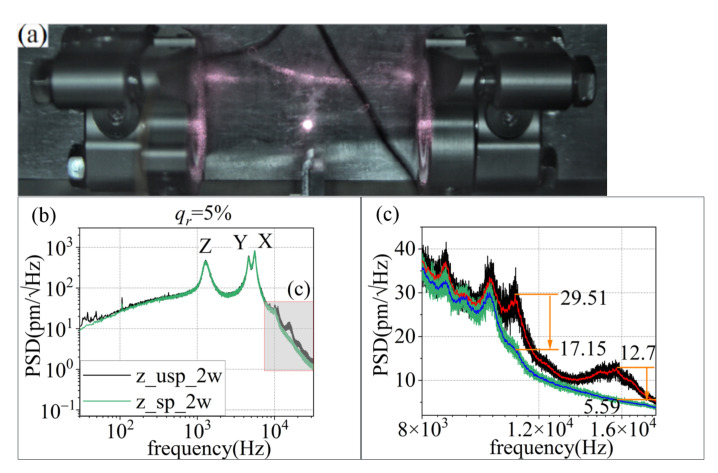
Suppression results of the PSD of center-of-mass displacement. (**a**) The captured particle. (**b**) The PSD after and before the 16.9 dB RIN suppression in the *Z*-axis. (**c**) Larger views of the areas of b.

**Figure 9 micromachines-13-00984-f009:**
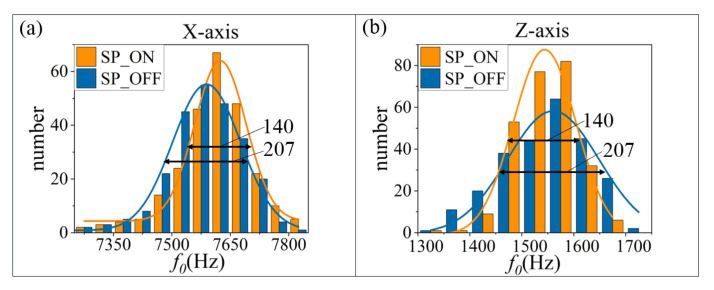
Statistical distribution of the resonant frequency. (**a**,**b**) are the Gaussian distribution of the resonant frequency along the *X* and *Z* axes, respectively.

## Data Availability

Some or all data, models, or code generated or used during the study are available from the corresponding author by request.

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
