# Peer review of "Analysis and Suppression of Laser Intensity Fluctuation in a Dual-Beam Optical Levitation System"

_micromachines, 2022, doi:10.3390/mi13070984_

Round 1

Reviewer 1 Report

This manuscript present the approach to analyze the effect of laser intensity fluctuation in dual-beam optical levitation system. It is a solid work that can be interesting and useful to the optical manipulation field. The presentation is very straightforward and I am happy to see its publication.

The minor suggestion I have is (1) maybe the authors can study the effect of particle size to confirm their method and (2) the authors are suggested to discuss the broad impact of this research and the potential applications.

Reviewer 2 Report

Present paper contains interesting and actual study of particle levitation stability inside dual-beam optical trap. Author used external control of laser power with AOM in order to improve the particle’s position detection sensitivity and the stability of the relaxation oscillation.

Authors established an analytical model to characterize the influence of laser intensity fluctuation in dual-beam optical trap. The results of measurements are consistent with the model prediction.
However, the quality of writing and description of the study is not high enough. Many important experimental and theoretical details are missed in the paper, such as the description of experimental realization of the power control system and the method of numerical solution of particle motion equation. In my opinion paper can be accepted only after major revision. Authors should explain their methods and results more clearly and consistently.

Point-by-point response:

11)      Line 146: “deal beam optical trap” should be changed to “dual beam optical trap”

22)      In the line 133 authors wrote “The power spectral density (PSD) is
calculated from the center-of-mass displacement x .” However, for me it was not clear how center-of-mass displacement was obtained. Was the center of mass displacement obtained using numerical solution of Equation 3? What kind of method was used to solve the equation? Authors should provide this information.

33)      The wavelength and power of the laser is not provided in the description of setup. It is provided later, together with the results. I found it very confusing and ask to place the information from lines 200-204 to setup description at lines 174-179.

44)      In the description of Fig. 6 authors should provide the transcript of all abbreviations, which are written in the figure

55)      In the line 176 authors wrote “The numerical apertures of both beams are approximately 0.07.” However, the value of 0.07 is low and it is not possible to get focal beam waste of 4 micrometers using the objective with such a low NA, it should be at least 2 times higher. Author should clarify this point.

66)      Authors wrote, that they used AOM to control the power of laser beams and thus to suppress the power fluctuation instability of trapping. However, I didn’t find any AOM in schematic on optical setup. Author should show where in the setup AOM was placed.

77)      Experimental realization of power stability control is not discussed in the paper at all. Only one sentence at line 200 “The built dual-beam optical trap contains intensity controls.” Development of the system of laser power control is one of the main results in this study and it should be discussed in details.

88)      It was not clear for me what kind of measured signal authors used to control AOM. Was it some signal from one of PDB? Or some additional measurements were performed? Authors should clarify this point in the paper

Round 2

Reviewer 2 Report

I believe the authors have sufficiently responded to the reviewers' comments